# Associations between HIV Status, SARS-CoV-2 Infection, Increase in Use of Psychoactive Substances and Oral Ulcers among People Who Used Psychoactive Substances during the First Wave of the COVID-19 Pandemic

**Morenike Oluwatoyin Folayan** [1,2,3,*], **Roberto Ariel Abeldaño Zuñiga** [1,4], **Jorma I. Virtanen** [1,5], **Oliver C. Ezechi** [1,6], **Nourhan M. Aly** [1,7], **Joanne Lusher** [1,8], **Annie L. Nguyen** [1,9] **and Maha El Tantawi** [1,7]

1 MEHEWE Study Group, Obafemi Awolowo University, Ile-Ife 220282, Nigeria
2 Department of Child Dental Health, Obafemi Awolowo University, Ile-Ife 220282, Nigeria
3 Oral Health Initiative, Nigerian Institute of Medical Research, Yaba, Lagos 101245, Nigeria
4 Postgraduate Department, University of Sierra Sur, Oaxaca 70800, Mexico
5 Faculty of Medicine, University of Bergen, 5020 Bergen, Norway
6 The Centre for Reproductive and Population Health Studies, Nigerian Institute of Medical Research, Yaba, Lagos 101245, Nigeria
7 Department of Pediatric Dentistry and Dental Public Health, Faculty of Dentistry, Alexandria University, Alexandria 5424041, Egypt
8 Provost's Group, Regent's University London, London NW1 4NS, UK
9 Department of Family Medicine, Keck School of Medicine, University of Southern California, Los Angeles, CA 90007, USA
* Correspondence: toyinukpong@yahoo.co.uk

**Abstract:** The aim of this study was to assess the associations between HIV status, SARS-CoV-2 infection, increase in use of psychoactive substances and oral ulcers among people who use psychoactive substances. This was a secondary analysis of the data of 1087 people who used psychoactive substances collected during the first wave of the COVID-19 pandemic. The data extracted were confounding (age, sex, the highest level of education attained, employment status, emotional distress status), dependent (oral ulcers) and independent (SARS-CoV-2 infection, increase in alcohol consumption, smoking and use of other psychoactive substances, living with HIV) variables. A multivariate logistic regression model was constructed to determine the associations between the dependent and independent variables after adjusting for the confounding variables. Participants who had a history of SARS-CoV-2 infection (AOR:10.37) and people living with HIV (AOR:1.91) had higher odds of reporting oral ulcers. The finding suggests that people who used psychoactive substances, had COVID-19 and lived with HIV were at increased risk for oral ulcers during the first wave of the COVID-19 pandemic. Increased use of psychoactive substances was not associated with a significant increase in the risk for oral ulcers. Further research is needed to better understand the reasons for these findings.

**Keywords:** alcohol consumption; cigarette smoking; psychoactive substances; stress-induced; global oral health; association; HIV; SARS-CoV-2; substance use; oral ulcers; COVID-19; pandemic

## 1. Introduction

Reports of the oral features of COVID-19 have emerged rapidly and include aphthous, hemorrhagic and necrotic ulcers [1]. In addition, other oral ulcers associated with COVID-19 are bullae, erythema multiforme-like lesions, macules, maculopapular, enanthema and pustules [2]. These ulcers may be virally induced through SARS-CoV-2 infection of the oral tissues. SARS-CoV-2 can be detected in saliva and oropharyngeal secretions and may express its pathogenic mechanisms in the oral and oropharyngeal mucosae [3]. These lesions may affect multiple oral sites in patients diagnosed with COVID-19 [3].

The incidence of oral ulcers may be heightened for people who are at risk for stress-induced oral ulcers. Stress-induced oral ulcers result from the increased level of salivary cortisol or reactive oxygen species and increases in the quantity and activity of leukocytes, oral tissue biting or injury to the oral mucosa due to aggravated actions [4]. COVID-19 induces physical and emotional stress [5]. Stress resulting from experiences of anxiety, anger, grief/sense of loss and sleep changes during the COVID-19 pandemic is associated with a high risk for oral ulcers [6].

The pandemic acted as an uncontrollable stress as the duration, intensity and impact on individuals were new and unknown. This form of stress lowers the sense of adaptability to the situation, increases the magnitude of the stress response and increases the risk for persistent homeostatic dysregulation associated with stress [7]. The brain regions associated with the perception and appraisal of emotional and stressful stimuli stimulate neural processes that evaluate the demands and assess the availability of adaptive resources to cope with the demands, as well as assess the behavioral, cognitive and physiological adaptation in response to stressors [8]. These assessments may lead to the initiation or escalation of the self-administration of nicotine, psychostimulants and alcohol [8].

Stress can increase the use of psychoactive substances [9], and prior studies have indicated that COVID-19-induced stress increases the use of psychoactive substances [10–12]. Ulcers are associated with the use of psychoactive substances [13,14]. The use of psychoactive substances causes xerostomia [15], and xerostomia is a cause of oral ulcers [16].

In addition, people living with HIV are at increased risk of having oral ulcers [17–19]. This results from immune system changes characterized by reversed CD4:CD8, lowered CD4 cell counts, an inverse correlation between CD4 cell counts and percent activated gamma delta lymphocytes and increased TNF alpha production in peripheral blood lymphocytes [20]. These immune alterations are associated with immune complex vasculitis causing oral ulcers [21]. In addition, the COVID-19 pandemic may disrupt access to and the use of antiretroviral drugs by people living with HIV [22] and thus further increase the risk of people living with HIV for oral ulcers.

Although the most common oral feature of COVID-19 is oral ulceration [3,23–32], little is currently known about how COVID-19 increases the risks of individuals who may have increased risk for stress to oral ulcers. Having this knowledge may assist with making prompt diagnosis for cases with a high index of suspicion and planning for targeted interventions for such populations. Thus, the aim of the present study was to determine associations between HIV status, COVID-19 status, increase in the use of psychoactive substances and oral ulcers among people who were using psychoactive substances during the first wave of the COVID-19 pandemic. We hypothesize that HIV positivity status, COVID-19 and an increase in the use of psychoactive substances will be associated with oral ulcers.

## 2. Materials and Methods

This study was a secondary analysis of a dataset of 21,206 participants in a study on the impact of COVID-19 mental health and wellness study. The data were collected from 152 countries between July and December 2020 using a questionnaire validated for global use [31]. The overall Content Validity Index of the questionnaire was 0.83. The methodology for this study had been comprehensively reported in a manuscript that comprehensively described the study methodology [31].

Ethical approval was obtained for this study from the Human Research Ethics Committee at the Institute of Public Health of the Obafemi Awolowo University Ile-Ife, Nigeria (HREC No: IPHOAU/12/1557). Additional ethical approval was obtained from India (D-1791-uz and D-1790-uz), Saudi Arabia (CODJU-2006F), Brazil (CAAE N° 38423820.2.0000.0010) and the United Kingdom (13283/10570). Study participants provided consent before participating in the online survey

Sample size

The data of 1087 participants who indicated they consumed alcohol or used tobacco or other psychoactive drugs at the time of the study were extracted for this analysis. The data of the sample extracted had a minimum of 10 participants with complete responses for each of the dependent variables for the study. This was considered statistically adequate for the performance of regression analyses with a minimum probability level of 0.05 [32]. The extracted data were those of study participants with complete responses to the dependent variable [33].

Participants Recruitment

Participants 18 years and above were recruited using an online survey tool (Survey Monkey®), the link to which was posted on Facebook, Twitter and Instagram; network email lists; and WhatsApp groups. Participants had to understand the languages the survey was offered in (English, French, Arabic, Portuguese or Spanish) and be able to access the survey using an electronic device connected to the internet. Respondents were encouraged to further share the link with their networks. Details on study participants' recruitment process was published previously [31,34].

Study Procedures

The survey took an average of 11 min to complete. It was preceded by an introduction of the purpose of the study and assurance of confidentiality and the voluntariness of participation. The check of a box indicating consent was required to be able to proceed with the survey. Each participant could only complete a single questionnaire because of the installed IP address restrictions. Participants could, however, edit their answers freely until they chose to submit. Full details of the methodology can be found elsewhere [31,35].

Dependent Variable

Participants were asked if they had had oral ulcers during the lockdown, with the response options of 'yes' or 'no'.

Independent Variables

*SARS-CoV-2 infection*: Respondents were asked if they had tested positive for COVID-19 during the study period with the response options of 'yes' or 'no'.

*Alcohol consumption, tobacco use and use of other psychoactive substances*: Participants were asked if they had experienced a change in alcohol consumption, tobacco use and use of other psychoactive substances during the pandemic (increase, started consuming or using, decrease, no change). The responses to each of the three questions were dichotomized into yes (increased/started using) and no (decreased/no change).

*Living with HIV*: Participants were asked about their HIV status by identifying if they were negative, positive, had unknown HIV status or were unwilling to declare their HIV status. Only data of respondents who were able to identify their HIV status were extracted. Responses were dichotomized into yes (HIV-positive) and no (HIV-negative).

Confounding Variables

*Sociodemographic variables*: The confounding sociodemographic variables were age in years, sex at birth (male, female, others), the highest level of education attained (none, primary, secondary, college/university) and employment status (employed, unemployed, student, retiree).

*Emotional distress*: Respondents were required to tick a checkbox for any of the listed emotions experienced during the pandemic. The list included depression, anxiety, frustration or boredom, loneliness, anger, grief or a feeling of loss. A ticked checkbox indicated that the participant was emotionally distressed. Responses were dichotomized to emotional distress present (yes) or emotional distress absent (no).

Data Analyses

Raw data were downloaded, cleaned and imported to SPSS version 23.0 (IBM SPSS Statistics for Windows, Version 23.0. Armonk, NY, USA: IBM Corp.) for analysis. Inferential analyses were conducted using multivariate logistic regression analysis to determine the associations between the dependent and independent variables while adjusting for confounding variables. Adjusted odds ratios (AoR) for the multivariate logistic regression

model and 95% confidence intervals (CI) were calculated. Statistical significance was set at <0.05.

## 3. Results

As shown in Table 1, of the 1087 participants whose data were extracted, 346 (31.8%) had oral ulcers, 148 (13.6%) had a SARS-CoV-2 infection during the first wave of the COVID-19 pandemic, 67 (6.2%) self-reported living with HIV, 210 (19.3%) increased or started consumption of alcohol during the pandemic, 206 (19.0%) increased or started smoking during the pandemic and 147 (13.5%) increased or started using other psychoactive substances during the pandemic.

**Table 1.** Multivariate logistic regression analysis to determine the association between COVID-19 status, HIV status, changes in alcohol consumption, tobacco use and use of other psychoactive substance and oral ulcers during the first wave of the COVID-19 pandemic [N = 1087].

| Variables | | Oral Ulcers | | AoR; 95% CI (*p* Values) |
|---|---|---|---|---|
| Age | Total N = 1087 (100%) | Yes N = 346 (31.8) n (%) | No N = 741 (68.2) n (%) | |
| **Sex at birth** | | | | |
| Male | 545 (50.1) | 197 (36.1) | 348 (63.9) | 3.856; 0.926–16.062; *p* = 0.064 |
| Female | 527 (48.5) | 144 (27.3) | 383 (383) | 3.100; 0.741–12.965; *p* = 0.121 |
| Others | 15 (1.4) | 5 (33.3) | 10 (66.7) | 1.000 |
| **Educational level** | | | | |
| No formal education | 22 (2.0) | 11 (50.0) | 11 (50.0) | 1.800; 0.682–4.755; *p* = 0.235 |
| Primary | 64 (5.9) | 53 (82.8) | 11 (17.2) | 15.388; 7.602–31.248; *p* < 0.001 |
| Secondary | 191 (17.6) | 49 (25.7) | 142 (74.3) | 1.511; 1.003–2.276; *p* = 0.049 |
| College/university | 810 (74.5) | 233 (29.8) | 577 (71.2) | 1.000 |
| **Employment status** | | | | |
| Retiree | 19 (1.7) | 3 (15.8) | 16 (84.2) | 0.674; 0.148–3.060; *p* = 0.609 |
| Student | 209 (19.2) | 36 (17.2) | 173 (82.8) | 1.221; 0.606–2.461; *p* = 0.576 |
| Employed | 757 (69.5) | 288 (38.0) | 469 (62.0) | 2.442; 1.345–4.434; *p* = 0.003 |
| Unemployed | 102 (9.4) | 19 (18.6) | 83 (81.4) | 1.000 |
| **Emotional distress** | | | | |
| Yes | 560 (51.5) | 133 (23.8) | 427 (76.3) | 0.590; 0.435–0.800; *p* = 0.001 |
| No | 527 (48.5) | 213 (40.3) | 314 (59.6) | 1.000 |
| **SARS-CoV-2 infection** | | | | |
| Yes | 148 (13.6) | 113 (76.4) | 35 (23.6) | 10.372; 6.615–16.263; *p* < 0.001 |
| No | 939 (86.4) | 233 (24.8) | 706 (75.2) | 1.000 |
| **Increased alcohol consumption** | | | | |
| Yes | 210 (19.3) | 76 (36.2) | 134 (63.8) | 0.898; 0.565–1.428; *p* = 0.649 |
| No | 877 (80.7) | 270 (30.8) | 607 (69.2) | 1.000 |
| **Increased smoking** | | | | |
| Yes | 206 (19.0) | 80 (38.8) | 126 (61.2) | 1.350; 0.861–2.118; *p* = 0.191 |
| No | 881 (81.0) | 266 (30.2) | 615 (69.8) | 1.000 |
| **Increased use of other psychoactive substance** | | | | |
| Yes | 147 (13.5) | 67 (45.6) | 80 (54.4) | 1.176; 0.717–1.926; *p* = 0.521 |
| No | 940 (86.5) | 279 (29.7) | 661 (70.3) | 1.000 |
| **Living with HIV** | | | | |
| Yes | 67 (6.2) | 35 (52.2) | 32 (47.8) | 1.912; 1.061–3.447; *p* = 0.031 |
| No | 1020 (93.8) | 311 (30.5) | 709 (69.5) | 1.000 |

Participants who had a history of SARS-CoV-2 infection had significantly higher odds of reporting oral ulcers during the first wave of the COVID-19 pandemic than those who did not have the infection (AOR: 10.37; 6.62–16.26; $p < 0.001$). In addition, people living with HIV had significantly higher odds of reporting oral ulcers than people not living with HIV during the first wave of the pandemic (1.91; 1.06–3.45; $p = 0.031$). Increases in alcohol consumption, tobacco use and use of other psychoactive substances were not significantly associated with increased risk for oral ulcers during the first wave of the pandemic.

## 4. Discussion

The present study attempted to identify sub-populations of people who use psychoactive substance with a risk for oral ulcers during the COVID-19 pandemic. This is the first study to explore these associations among this high-risk group that we are aware of. Our findings suggest that people who used psychoactive substances during the COVID-19 pandemic and who also had a SARS-CoV-2 infection had over a 10-fold increase in risk for oral ulcers. In addition, those living with HIV had increased risk for oral ulcers. An increase in the use of psychoactive substances (alcohol, tobacco and other psychoactive substances) did not significantly increase the risk for oral ulcers during the first wave of the COVID-19 pandemic. The study findings partly support our hypothesis that HIV positivity status, COVID-19 infection and an increase in the use of psychoactive substances is associated with oral ulcers.

One of the strengths of the current study is the population-specific analysis conducted on the risk for oral ulcers during the COVID-19 pandemic. Despite this study generating some thought-provoking findings, there are, however, some limitations that should be highlighted. First, this was a cross-sectional survey, which limits the possibility of making causal inferences. Data were also generated through online participation with the risk of excluding people with low socioeconomic status who are not able to have access to a smartphone or the Internet. The study thus ran the risk of excluding a population who are more likely to experience stress and oral ulcers, as the association between the dependent and the confounding variable (educational status) suggests. However, the collection of online data was the most appropriate strategy during the COVID-19 pandemic. The public health measures instituted to reduce the risk of contracting the disease limited our options. These issues modestly limit the generalizability of study findings. In addition, a number of constructs were subjectively rather than objectively measured using single-item questions. Single-item questions are, however, considered to be good measures of phenomena, are as effective as multi-item tests and are far more time-efficient once they are valid questions [36]. Despite these limitations, the study generated important findings.

First, we observed that exposure to viral infections—SARS-CoV-2 or HIV—increased the risk for oral ulcers. The direct viral infection of tissues of the oral cavity may cause cellular destruction or immune reaction to the viral proteins, thereby causing ulceration of the oral tissue [37]. However, the risk for oral ulcers from SARS-CoV-2 infection was significantly higher than it was from HIV infection. The experience of stress *per se*, as indicated by the increase in use of psychoactive substances during the pandemic, may not be an absolute risk factor for oral ulcers during the COVID-19 pandemic. Prior studies had suggested that the interaction between SARS-CoV-2 and angiotensin converting enzyme 2 (ACE2) might disrupt the function of oral keratinocytes and the epithelial lining of salivary glands ducts, resulting in painful oral ulcers [38]. The presence of ACE2 receptors on the tongue and salivary glands, the affinity of SARS-CoV-2 for ACE2 and the affectation of the tongue and of the salivary glands (dysgeusia) by SARS-CoV-2 infection may potentially allow for oral mucosal ulcerations and necrosis to result directly from SARS-CoV-2 infection [38]. There is, however, no clarity about whether the oral ulcers are a direct result of infection with the virus or if they are secondary to the compromised state of patients with SARS-CoV-2 [39]. The findings from the current study suggest that studies on the pathophysiology of oral ulcers resulting from direct infection of the oral mucosa may be the more viable route for the formation of oral ulcers associated with COVID-19 infection.

Second, we observed that people living with HIV who use psychoactive substances may have increased risk for oral ulcers, though increased use of psychoactive substances may not significantly increase the risk for oral ulcers. These findings suggests that oral ulcers associated with HIV infection may be mediated through the use of psychoactive substances and not mediated by stress (proximally measured by the increased use of psychoactive substances during the COVID-19 pandemic). It is therefore plausible that immunochemical actions resulting from viral infection of the oral tissues mediated by the use of psychoactive substances increases the risk of people living with HIV to oral ulcers during the COVID-19 pandemic. Perhaps synergistic interactions exist between the virus, which increases the risk for oral ulcers, and the chemicals from the psychoactive substances. In fact, one author suggests the possibility of 'crosstalk of immune-mediated pathways' underlying the pathogenesis of oral ulcers associated with SARS-CoV-2 infection [30]. These postulations remain speculative and warrant further investigation

Third we observed that increases in the use of psychoactive substances (alcohol, tobacco, other psychoactive substances) was not significantly associated with an increase in risk for oral ulcers. Although the use of psychoactive substances is associated with oral ulcers [11,12] and the increase in use of psychoactive substances during the COVID-19 pandemic was indicative of greater stress [40], this link between the increase in use of psychoactive substances and stress does not seem to translate to increased risk for oral ulcers during the COVID-19 pandemic. This might be because the threshold at which increases in stress leading to increased use of psychoactive substances occurs may not have been elevated to a level that leads to a significant increase in the risk for oral ulcers. Further work is necessary to unravel this intricacy.

Our postulation is that viral infections caused by HIV and SARS-CoV-2 result in oral ulcers due to infection of the oral tissues. Psychoactive substances enhance the interactions between viruses and the cellular tissues, and this interaction is not dependent on the frequency or intensity of use of psychoactive substances. The impact of SARS-CoV-2 on cellular responses that results in oral ulcer formation seems to be more intense than that of HIV, suggesting that there may be differences in the impact of acute and chronic viral infection of the oral tissues leading to oral ulcer formation. Stress may not play a role in moderating the interactions between psychoactive substance, viruses and the oral tissue cellular responses that cause oral ulcers, though stress can independently include oral ulcer formation. Studies are needed to explore these hypotheses.

Our study findings may also assist with making prompt diagnoses for cases with a high index of suspicion and in planning for targeted interventions for such populations. We suggest that a diagnosis of oral ulcers during the COVID-19 pandemic should raise a high index of suspicion for SARs-CoV-2 infection. Management should include appropriate management for viral infections, including the prescription of antivirals targeted at SARS-CoV-2.

## 5. Conclusions

Findings from the present study suggest that people who used psychoactive substances and who were positive for COVID-19 and HIV were at increased risk for oral ulcers during the first wave of the COVID-19 pandemic. However, increased use of psychoactive substances during the first wave of the COVID-19 pandemic was not associated with a significant increase in the risk for oral ulcers. These findings offer a unique contribution to our current understanding on the pathophysiology of oral ulcers and suggest that the oral ulcers observed during the COVID-19 pandemic may be a result of infection of the oral tissue by the virus rather than being stress-induced. This needs to be studied further.

**Author Contributions:** Conceptualization, M.O.F.; methodology, M.O.F., M.E.T., O.C.E. and A.L.N.; formal analysis, R.A.A.Z.; investigation, M.O.F., R.A.A.Z., J.I.V., O.C.E., N.M.A., J.L., M.E.T. and A.L.N.; data curation, M.O.F.; R.A.A.Z., J.I.V., O.C.E., N.M.A., J.L., M.E.T. and A.L.N.; writing—original draft preparation, M.O.F.; writing—review and editing, M.O.F., R.A.A.Z., J.I.V., O.C.E., N.M.A., J.L., M.E.T. and A.L.N.; supervision, M.O.F., O.C.E., N.M.A., M.E.T. and A.L.N.; project administration, M.O.F., O.C.E., N.M.A., M.E.T. and A.L.N. All authors have read and agreed to the published version of the manuscript.

**Funding:** This research received no external funding.

**Institutional Review Board Statement:** Ethical approval of the current study was obtained from the Human Research Ethics Committee at Institute of Public Health of the Obafemi Awolowo University Ile-Ife, Nigeria (HREC No: IPHOAU/12/1557) as the lead partner for this study. The protocol was in accordance with international and national research guidelines. All participants provided written informed consent before taking the survey.

**Informed Consent Statement:** Informed consent was obtained from all subjects involved in the study.

**Data Availability Statement:** Available upon request.

**Acknowledgments:** The authors wish to thank all those who participated in this study.

**Conflicts of Interest:** The authors declare no conflict of interest.

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
