# Peer review of "Associations between HIV Status, SARS-CoV-2 Infection, Increase in Use of Psychoactive Substances and Oral Ulcers among People Who Used Psychoactive Substances during the First Wave of the COVID-19 Pandemic"

_2673-947X, doi:10.3390/hygiene3020009_

Round 1

Reviewer 1 Report

I found the article titled "Associations between HIV status, SARS-CoV-2 infection, in-2 crease in use of psychoactive substances and oral ulcers 3 among people who used psychoactive substances during the 4 first wave of the COVID-19 pandemic" interesting and scientifically sound.

Some minor corrections are suggested as follows:

Line 140-141. Please correct this line.

Line 221. use COVID-19 instead COVID19.

Line 227. Please use capital D.

Line 269, 272. Please use SARS-CoV-2 instead of SARS-C0V-2.

Please prepare the reference list using the MDPI style.

Author Response

I found the article titled "Associations between HIV status, SARS-CoV-2 infection, in-2 crease in use of psychoactive substances and oral ulcers 3 among people who used psychoactive substances during the 4 first wave of the COVID-19 pandemic" interesting and scientifically sound.

Response: Thanks for the feedback

Some minor corrections are suggested as follows:

Line 140-141. Please correct this line.

Response: edit effected

Line 221. use COVID-19 instead COVID19.

Response: edit effected

Line 227. Please use capital D.

Response: edit effected

Line 269, 272. Please use SARS-CoV-2 instead of SARS-C0V-2.

Response: edit effected

Please prepare the reference list using the MDPI style.

Response: Thanks a million for this observation. The journal usually supports with this process when processing the manuscript. we will rely on this support

Reviewer 2 Report

Thank you very much for this interesting paper regarding oral disease during the first wave of the SARS-CoV2 pandemic situation.

Regarding language and formal points some adjustments are needed and minor corrections should be addressed.

Introduction: Please give a clear hypothesis and null hypothesis to be tested. Please explain the possible correlation between oral expressions of medical factors like infection with viruses more in detail. 

Material and Methods: A control group without alcohol intake was not evaluated in the study. Why. Please add or explain.

Discussion: Please give clinical recommendations basing on your results. How to treat or avoid the presented oral pathologies. This might increase the benefit of your paper for the dental community.

Author Response

Thank you very much for this interesting paper regarding oral disease during the first wave of the SARS-CoV2 pandemic situation.

Response: Thanks for the feedback

Regarding language and formal points some adjustments are needed and minor corrections should be addressed.

Response: Thanks for the feedback. We have worked on this

Introduction: Please give a clear hypothesis and null hypothesis to be tested. Please explain the possible correlation between oral expressions of medical factors like infection with viruses more in detail. 

Response: Thanks for raising this. We wrote: We hypothesize that HIV status, COVID-19 status and increase in use of psychoactive substances will be associated with oral ulcers.

Material and Methods: A control group without alcohol intake was not evaluated in the study. Why. Please add or explain.

Response:  Thanks a million for raising this question. The conceptual framework for this study suggests that increase in alcohol use (and not use vs none-use) may be associated with oral ulcers. We therefore tested this concept in this study.

Discussion: Please give clinical recommendations basing on your results. How to treat or avoid the presented oral pathologies. This might increase the benefit of your paper for the dental community.

Response: thanks for this recommendation. We have included a suggestion on patients management in the discussion. We wrote: Our study findings may also assist with making prompt diagnosis for cases with high index of suspicion and planning for targeted interventions for such populations. We suggest that a diagnosis of oral ulcers during the COVID-19 pandemic should raise a high index of suspicion for SARs-CoV-2 infection. Management should include appropriate management for viral infections including prescription of anitivirals targeted at SARS-CoV-2.